# RingAttention with Blockwise Transformers for Near-Infinite Context

**Hao Liu, Matei Zaharia, Pieter Abbeel**
UC Berkeley

## Abstract

Transformers have emerged as the architecture of choice for many state-of-the-art AI models, showcasing exceptional performance across a wide range of AI applications. However, the memory demands imposed by Transformers limit their ability to handle long sequences, thereby posing challenges in utilizing videos, actions, and other long-form sequences and modalities in complex environments. We present a novel approach, Blockwise RingAttention, which leverages blockwise computation of self-attention and feedforward to distribute long sequences across multiple devices while fully overlapping the communication of key-value blocks with the computation of blockwise attention. Our approach enables training and inference of sequences that are up to device count times longer than those achievable by prior memory-efficient Transformers, without resorting to approximations or incurring additional communication and computation overheads. Extensive experiments on language modeling and reinforcement learning tasks demonstrate the effectiveness of our approach in allowing millions of tokens context size and improving performance.

## 1 Introduction

Transformers (Vaswani et al., 2017) have become the backbone of many state-of-the-art AI systems that have demonstrated impressive performance across a wide range of AI problems. Transformers achieve this success through their architecture design that uses self-attention and position-wise feedforward mechanisms. However, scaling up the context length of Transformers is a challenge (OpenAI, 2023), since the inherited architecture design of Transformers, *i.e.* the self-attention has memory cost quadratic in the input sequence length, which makes it challenging to scale to longer input sequences. Large context Transformers are essential for tackling a diverse array of AI challenges, ranging from processing books and high-resolution images to analyzing long videos and complex codebases. They excel at extracting information from the interconnected web and hyperlinked content, and are crucial for handling complex scientific experiment data. There have been emerging use cases of language models with significantly expanded context than before: GPT-3.5 (Schulman et al., 2022) with context length 16K, GPT-4 (OpenAI, 2023) with context length 32k, MosaicML's MPT (MosaicML, 2023) with context length 65k, and Anthropic's Claude (Anthropic, 2023) with context length 100k.

Driven by the significance, there has been surging research interests in reducing memory cost. One line of research leverages the observation that the softmax matrix in self-attention can be computed without materializing the full matrix (Milakov and Gimelshein, 2018) which has led to the development of blockwise computation of self-attention and feedforward (Rabe and Staats, 2021; Dao et al., 2022; Liu and Abbeel, 2023b) without making approximations. Despite the reduced memory, a significant challenge still arises from storing the output of each layer. This necessity arises from self-attention's inherent nature, involving interactions among all elements (n to n interactions). The subsequent layer's self-attention relies on accessing all of the prior layer's outputs. Fail-

---

Code is available. Email: hao.liu@cs.berkeley.edu

ing to do so would increase computational costs cubically, as every output must be recomputed for each sequence element, rendering it impractical for longer sequences.

These components facilitate the efficient capture of long-range dependencies between input tokens, and enable scalability through highly parallel computations. To put the memory demand in perspective, even when dealing with a batch size of 1, processing 100 million tokens requires over 1000GB of memory for a modest model with a hidden size of 1024. This is much greater than the capacity of contemporary GPUs and TPUs, which typically have less than 100GB of high-bandwidth memory (HBM).

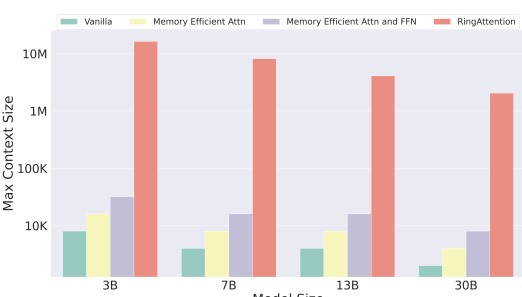

**Figure 1** Maximum context length under end-to-end large-scale training on TPUv4-1024. Baselines are vanilla transformers (Vaswani et al., 2017), memory efficient transformers (Rabe and Staats, 2021), and memory efficient attention and feedforward (blockwise parallel transformers) (Liu and Abbeel, 2023b). Our proposed approach Blockwise RingAttention allows training up to device count times longer sequence than baselines and enables the training of sequences that exceed millions in length without making approximations nor adding any overheads to communication and computation.

To tackle this challenge, we make a key observation: by performing self-attention and feedforward network computations in a blockwise fashion (Liu and Abbeel, 2023b), we can distribute sequence dimensions across multiple devices, allowing concurrent computation and communication. This insight stems from the fact that when we compute the attention on a block-by-block basis, the results are invariant to the ordering of these blockwise computations. Our method distributes the outer loop of computing blockwise attention among hosts, with each device managing its respective input block. For the inner loop, every device computes blockwise attention and feedforward operations specific to its designated input block. Host devices form a conceptual ring, where during the inner loop, each device sends a copy of its key-value blocks being used for blockwise computation to the next device in the ring, while simultaneously receiving key-value blocks from the previous one. As long as block computations take longer than block transfers, overlapping these processes results in no added overhead compared to standard transformers. Our work utilizes blockwise parallel transformers (Liu and Abbeel, 2023b) to substantially reduce memory costs, enabling zero-overhead scaling of context size across tens of millions of tokens during both training and inference, and allowing for the use of an arbitrarily large context size. Since our approach overlaps the communication of key-value blocks between hosts in a ring through blockwise computation of transformers, we name it RingAttention with Blockwise Transformers.

We evaluate the effectiveness of our approach on language modeling benchmarks. Our experiments show that RingAttention can reduce the memory requirements of Transformers, enabling us to train more than 500 times longer sequence than prior memory efficient state-of-the-arts and enables the training of sequences that exceed 100 million in length without making approximations to attention. Importantly, RingAttention eliminates the memory constraints imposed by individual devices, empowering the training and inference of sequences with lengths that scale in proportion to the number of devices, essentially achieving near-infinite context size.

Our contributions are twofold: (a) proposing a memory efficient transformers architecture that allows the context length to scale linearly with the number of devices while maintaining performance, eliminating the memory bottleneck imposed by individual devices, and (b) demonstrating the effectiveness of our approach through extensive experiments.

## 2    LARGE CONTEXT MEMORY CONSTRAINT

Given input sequences $Q, K, V \in \mathbb{R}^{s \times d}$ where $s$ is the sequence length and $d$ is the head dimension. We compute the matrix of outputs as:

$$\text{Attention}(Q, K, V) = \text{softmax}(\frac{QK^T}{\sqrt{d}})V,$$

where softmax is applied row-wise. Each self-attention sub-layer is accompanied with a feedforward network, which is applied to each position separately and identically. This consists of two linear transformations with a ReLU activation in between.

$$\text{FFN}(x) = \max(0, xW_1 + b_1)W_2 + b_2.$$

**Blockwise Parallel Transformers.** Prior state-of-the-arts have led to substantial reductions in memory utilization, achieved through innovative techniques that enable attention computation without full materialization by computing attention in a block by block manner (Rabe and Staats, 2021; Dao et al., 2022; Liu and Abbeel, 2023b). These advancements lowered the memory overhead of attention to $2bsh$ bytes per layer, where $b$ represents the batch size, $s$ denotes the sequence length, and $h$ stands for the hidden size of the model. To further reduce memory usage, blockwise parallel transformer (BPT) (Liu and Abbeel, 2023b) introduced a strategy where the feedforward network associated with each self-attention sub-layer is computed in a block-wise fashion. This approach effectively limits the maximum activation size of feedforward network from $8bsh$ to $2bsh$. For a more detailed analysis of memory efficiency, please refer to the discussion provided therein. In summary, the state-of-the-art transformer layer's memory cost of activation is $2bsh$.

**Large Output of Each Layer.** While BPT significantly reduces memory demand in Transformers, it still presents a major challenge for scaling up context length because it requires storing the output of each layer. This storage is crucial due to the inherent nature of self-attention, which involves interactions among all elements (n to n interactions). Without these stored outputs, the subsequent layer's self-attention becomes computationally impractical, necessitating recomputation for each sequence element. To put it simply, processing 100 million tokens with a batch size of 1 requires over 1000GB of memory even for a modest model with a hidden size of 1024. In contrast, modern GPUs and TPUs typically provide less than 100GB of high-bandwidth memory (HBM), and the prospects for significant HBM expansion are hindered by physical limitations and high manufacturing costs.

## 3    RINGATTENTION WITH BLOCKWISE TRANSFORMERS

Our primary objective is to eliminates the memory constraints imposed by individual devices by efficiently distribute long sequences across multiple hosts without adding overhead. To achieve this goal, we propose an enhancement to the blockwise parallel transformers (BPT) framework (Liu and Abbeel, 2023b). When distributing an input sequence across different hosts, each host is responsible for running one element of the outer loop of blockwise attention corresponding to its designated block, as well as the feedforward network specific to that block. These operations do not necessitate communication with other hosts. However, a challenge arises in the inner loop, which involves key-value block interactions that require fetching blocks from other hosts. Since each host possesses only one key-value block, the naive approach of fetching blocks from other hosts results in two significant issues. Firstly, it introduces a computation delay as the system waits to receive the necessary key-value blocks. Secondly, the accumulation of key-value blocks leads to increased memory usage, which defeats the purpose of reducing memory cost.

**Ring-Based Blockwise Transformer.** To tackle the aforementioned challenges, we leverage the permutation invariance property of the inner loop's key-value block operations. This property stems from the fact that the self-attention between a query block and a group of key-value blocks can be computed in any order, as long as the statistics of each block are combined correctly for rescaling. We leverage this property by conceptualizing all hosts as forming a ring structure: host-1, host-2, ..., host-$N$. As we compute blockwise attention and

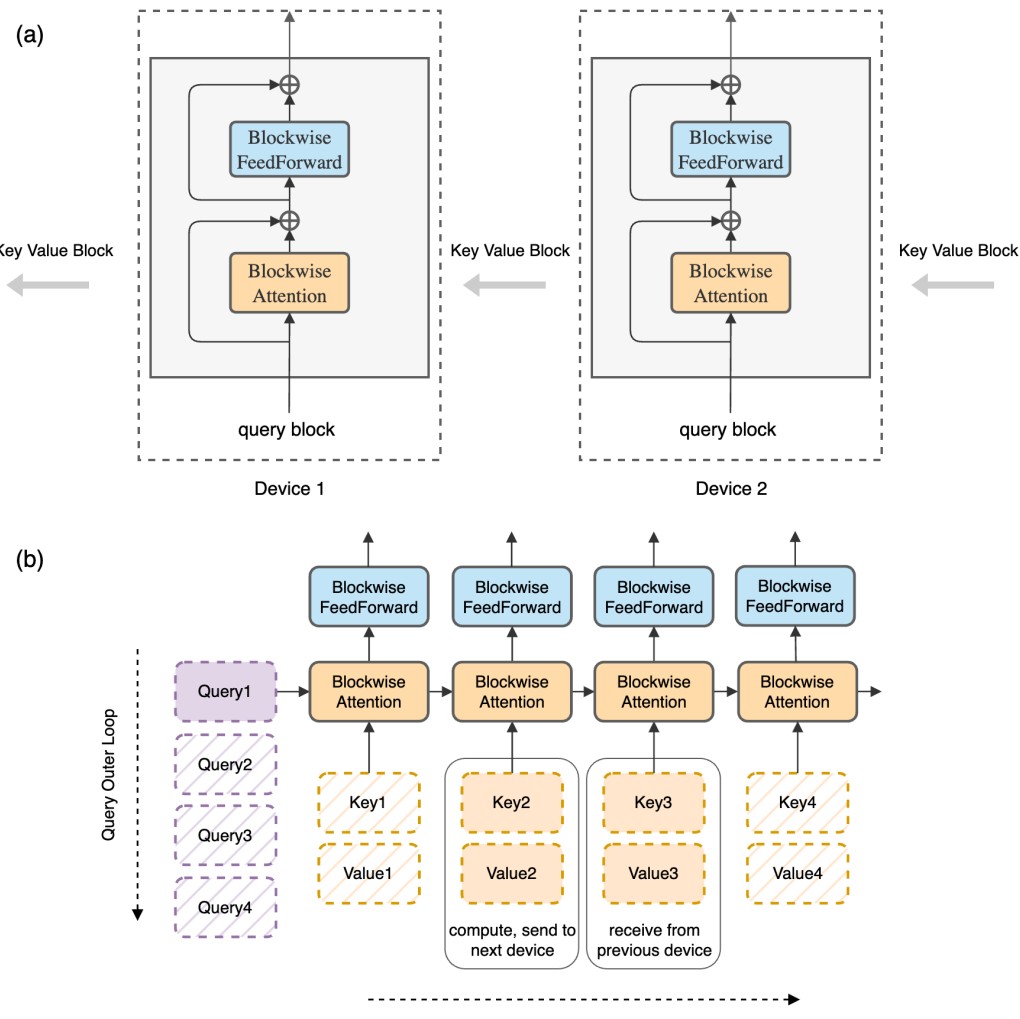

**Figure 2** **Top (a):** We use the same model architecture as the original Transformer but reorganize the compute. In the diagram, we explain this by showing that in a ring of hosts, each host holds one query block, and key-value blocks traverse through a ring of hosts for attention and feedforward computations in a block-by-block fashion. As we compute attention, each host sends key-value blocks to the next host while receives key-value blocks from the preceding host. The communication is overlapped with the computation of blockwise attention and feedforward. **Bottom (b):** We compute the original Transformer block-by-block. Each host is responsible for one iteration of the query's outer loop, while the key-value blocks rotate among the hosts. As visualized, a device starts with the first query block on the left; then we iterate over the key-value blocks sequence positioned horizontally. The query block, combined with the key-value blocks, are used to compute self-attention (yellow box), whose output is pass to feedforward network (cyan box).

feedforward, each host efficiently coordinates by concurrently sending key-value blocks being used for attention computation to the next host while receiving key-value blocks from the preceding host, effectively overlapping transferring of blocks with blockwise computation. Concretely, for any host-$i$, during the computation of attention between its query block and a key-value block, it concurrently sends key-value blocks to the next host-$(i+1)$ while receiving key-value blocks from the preceding host-$(i-1)$. If the computation time exceeds the time required for transferring key-value blocks, this results in no additional communication cost. This overlapping mechanism applies to both forward and backward passes of our approach since the same operations and techniques can be used. Prior work has also proposed leveraging a ring topology to compute full attention (Li et al., 2023b), aiming to reduce communication

**Table 1** Comparison of maximum activation sizes among different Transformer architectures. Here, $b$ is batch size, $h$ is hidden dimension, $n$ is number of head, $s$ is sequence length, $c$ is block size, the block size ($c$) is independent of the input sequence length ($s$). The comparison is between vanilla transformer Vaswani et al. (2017), memory efficient attention (Rabe and Staats, 2021; Dao et al., 2022), blockwise parallel transformers (Liu and Abbeel, 2023b), and our proposed approach RingAttention. Numbers are shown in bytes per layer, assuming *bfloat16* precision.

| Layer Type | Self-Attention | FeedForward | Total |
|---|---:|---:|---:|
| Vanilla Transformers | $2bns^2$ | $8bsh$ | $2bhs^2$ |
| Memory Efficient Attention | $2bsh + 4bch$ | $8bsh$ | $8bsh$ |
| Blockwise Parallel Transformers | $2bsh$ | $2bsh$ | $2bsh$ |
| Blockwise RingAttention | $6bch$ | $2bch$ | $6bch$ |

costs. Our work differs by utilizing blockwise parallel transformers to substantially reduce memory costs. As we show in the next section, this enables zero-overhead scaling of context size during both training and inference and allows arbitrarily large context size.

**Arithmetic Intensity Between Hosts.** In order to determine the minimal required block size to overlap transferring with computation, assume that each host has $F$ FLOPS and that the bandwidth between hosts is denoted as $B$. It's worth noting that our approach involves interactions only with the immediately previous and next hosts in a circular configuration, thus our analysis applies to both GPU all-to-all topology and TPU torus topology. Let's consider the variables: block size denoted as $c$ and hidden size as $d$. When computing blockwise self-attention, we require $2dc^2$ FLOPs for calculating attention scores using queries and keys, and an additional $2dc^2$ FLOPs for multiplying these attention scores by values. In total, the computation demands amount to $4dc^2$ FLOPs. We exclude the projection of queries, keys, and values, as well as blockwise feedforward operations, since they only add compute complexity without any communication costs between hosts. This simplification leads to more stringent condition and does not compromise the validity of our approach. On the communication front, both key and value blocks require a total of $2cd$ bytes. Thus, the combined communication demand is $4cd$ bytes. To achieve an overlap between communication and computation, the following condition must hold: $4dc^2/F \geq 4cd/B$. This implies that the block size, denoted as $c$, should be greater than or equal to $F/B$. Effectively, this means that the block size needs to be larger than the ratio of FLOPs over bandwidth.

**Memory Requirement.** A host needs to store multiple blocks, including one block size to store the current query block, two block sizes for the current key and value blocks, and two block sizes for receiving key and value blocks. Furthermore, storing the output of blockwise attention and feedforward necessitates one block size, as the output retains the shape of the query block. Therefore, a total of six blocks are required, which translates to $6bch$ bytes of memory. It's worth noting that the blockwise feedforward network has a maximum activation size of $2bch$ (Liu and Abbeel, 2023b). Consequently, the total maximum activation size remains at $6bch$ bytes. Table 1 provides a detailed comparison of the memory costs between our method and other approaches. Notably, our method exhibits the advantage of linear memory scaling with respect to the block size $c$, and is independent of the input sequence length $s$.

Our analysis shows that the model needs to have a sequence length of $s = 6c$, which is six times the minimal block size. Requirements for popular computing servers are shown in Table 2. The required minimal sequence length (rightmost column) for each host varies between 6K and 10K, and the minimal block size (second-to-rightmost column) for each host is around 1K for TPUs and GPUs with high bandwidth interconnect. For GPUs connected via InfiniBand, which offers lower bandwidth, the requirements are more strict. These requirements are easy to meet using blockwise transformers (Liu and Abbeel, 2023b) and standard parallelism such as data and tensor parallelism, which we will show in experiment Section 5.

**Table 2** Minimal sequence length needed on each device. Interconnect Bandwidth is the unidirectional bandwidth between hosts, *i.e.*, NVLink / InfiniBand bandwidth between GPUs and ICI bandwidth between TPUs. The minimal block size required $c = \text{FLOPS/Bandwidth}$, and minimal sequence length $s = 6c$.

| Spec Per Host | FLOPS | HBM | Interconnect Bandwidth | Minimal Blocksize | Minimal Sequence Len |
|---|---|---|---|---|---|
| | (TF) | (GB) | (GB/s) | ($\times$1e3) | ($\times$1e3) |
| A100 NVLink | 312 | 80 | 300 | 1.0 | 6.2 |
| A100 InfiniBand | 312 | 80 | 12.5 | 24.5 | 149.5 |
| TPU v3 | 123 | 16 | 112 | 1.1 | 6.6 |
| TPU v4 | 275 | 32 | 268 | 1.0 | 6.2 |
| TPU v5e | 196 | 16 | 186 | 1.1 | 6.3 |

---

**Algorithm 1** Large Context Transformers using RingAttention with Blockwise Transformers.

---

**Required:** Input sequence $x$. Number of hosts $N_h$.
Initialize
Split input sequence into $N_h$ blocks that each host has one input block.
Compute query, key, and value for its input block on each host.
**for** Each transformer layer **do**
    **for** $count = 1$ **to** $N_h - 1$ **do**
        **for** For each host concurrently. **do**
            Compute memory efficient attention incrementally using local query, key, value blocks.
            Send key and value blocks to next host and receive key and value blocks from previous host.
        **end for**
    **end for**
    **for** For each host concurrently. **do**
        Compute memory efficient feedforward using local attention output.
    **end for**
**end for**

---

**Algorithm and Implementation.** Algorithm 1 provides the pseudocode of the algorithm. RingAttention is compatible with existing code for memory efficient transformers: RingAttention just needs to call whatever available memory efficient computation locally on each host, and overlap the communication of key-value blocks between hosts with blockwise computation. We use collective operation `jax.lax.ppermute` to send and receive key value blocks between nearby hosts. A Jax implementation is provided in Appendix A.

## 4 SETTING

We evaluate the impact of using RingAttention in improving Transformer models by benchmarking maximum sequence length and model flops utilization.

**Model Configuration.** Our study is built upon the LLaMA architecture, we consider 3B, 7B, 13B, and 30B model sizes in our experiments.

**Baselines.** We evaluate our method by comparing it with vanilla transformers (Vaswani et al., 2017) which computes self-attention by materializing the attention matrix and computes the feedforward network normally, transformers with memory efficient attention (Rabe and Staats, 2021) and its efficient CUDA implementation (Dao et al., 2022), and transformers with both memory efficient attention and feedforward (Liu and Abbeel, 2023b).

**Training Configuration.** For all methods, we apply full gradient checkpointing (Chen et al., 2016) to both attention and feedforward, following prior works (Rabe and Staats, 2021; Liu and Abbeel, 2023b). The experiments are on both GPUs and TPUs. For GPUs,

we consider both single DGX A100 server with 8 GPUs and distributed 32 A100 GPUs. We also experiment with TPUs, from older generations TPUv3 to newer generations of TPUv4 and TPUv5e. We note that all of our results are obtained using full precision instead of mixed precision.

## 5 RESULTS

In our experiments, our primary objective is to comprehensively evaluate the performance of RingAttention across multiple key metrics, including maximum supported sequence length within accelerator memory, model flops utilization, and throughput. We compare RingAttention's performance with several baseline models , including the vanilla transformers (Vaswani et al., 2017), transformers with memory efficient attention (Rabe and Staats, 2021), and transformers with both memory efficient attention and feedforward (Liu and Abbeel, 2023b), across different model sizes and accelerator configurations.

### 5.1 EVALUATING MAX CONTEXT SIZE

We evaluate maximum supported context length using fully sharded tensor parallelsim (FSDP) (Facebook, 2023) which is widely used in prior end-to-end training (Touvron et al., 2023; Geng and Liu, 2023). We note that no tensor parallelism is considered in our evaluations since our approach is independent of tensor parallelism. Practitioners can combine our method with tensor parallelism, which we will show in Section 5.2. Using FSDP allows us to set the same batch size in tokens for baselines and our approach, ensuring a fair comparison. Concretely, on $n$ devices, FSDP is used to shard the model for baselines, which gives a sequence length of $l$. The total batch size in tokens is $nl$. We utilize FSDP along with RingAttention to extend the sequence length to $\frac{nl}{m}$ and $m$ sequences. This means that the total batch size in tokens remains the same, but RingAttention enables a significantly larger context size. Table 3 summarizes the results of our experiments.

Our RingAttention model consistently surpasses baselines, delivering superior scalability across diverse hardware setups. For example, with 32 A100 GPUs, we achieve over 1 million tokens in context size for 7B model, a 32 times improvement over previous best. Furthermore, when utilizing larger accelerators like TPUv4-512, RingAttention enables a 256 times increase in context size, allows training sequences of over 30 million tokens. Furthermore, our RingAttention model scales linearly with the number of devices, as demonstrated by the 8x improvement over previous best on 8 A100 and the 256x improvement on TPUv3-512. If a model can be trained with context size $s$ on $n$ GPUs using the blockwise attention and feedforward, with our RingAttention approach, it becomes possible to train a model with a context size of $ns$.

### 5.2 EVALUATING MODEL FLOPS UTILIZATION

We evaluate the model flops utilization (MFU) of RingAttention in standard training settings using fully sharded data parallelism(FSDP) (Facebook, 2023) and tensor parallelism following LLaMA and OpenLLaMA (Touvron et al., 2023; Geng and Liu, 2023) with Jax SPMD. The batch size in tokens are 2M on 8/32x A100 and 4M on TPUv4-256. Our goal is investigating the impact of model size and context length on MFU, a critical performance metrics while highlighting the benefits of our approach. Table 4 presents the results of our experiments on MFU for different model sizes and context lengths. We present the achieved MFU using state-of-the-art memory efficient transformers BPT (Liu and Abbeel, 2023b), compare it to our anticipated MFU based on these results, and demonstrate the actual MFU obtained with our approach (RingAttention). For fair comparison, both BPT and our approach are based on the same BPT implementation on both GPUs and TPUs.

RingAttention trains much longer context sizes for self-attention, resulting in higher self-attention FLOPs compared to baseline models. Since self-attention has a lower MFU than feedforward, RingAttention is expected to have a lower MFU than the baseline models. Our method offers a clear advantage in terms of maintaining MFU while enabling training with significantly longer context lengths. As shown in Table 4, when comparing our approach

**Table 3** The maximum context length supported in end-to-end training using fully sharded data parallelism and various transformers architectures. We show different model sizes and accelerators. Baselines are vanilla transformer (Vaswani et al., 2017), transformer with memory efficient attention (Rabe and Staats, 2021), and transformer with memory efficient attention and feedforward (Liu and Abbeel, 2023b). The context size is reported in tokens (1e3). Our approach Blockwise RingAttention substantially outperforms baselines and enables training sequences that are up to device count times longer than prior state-of-the-arts.

| | Max context size supported (×1e3) | | | | |
|---|---|---|---|---|---|
| | Vanilla | Memory Efficient Attn | Memory Efficient Attn and FFN | RingAttention (Ours) | Ours vs SOTA |
| 8x A100 NVLink | | | | | |
| 3B | 4 | 32 | 64 | **512** | 8x |
| 7B | 2 | 16 | 32 | **256** | 8x |
| 13B | 2 | 4 | 16 | **128** | 8x |
| 32x A100 InfiniBand | | | | | |
| 7B | 4 | 64 | 128 | **4096** | 32x |
| 13B | 4 | 32 | 64 | **2048** | 32x |
| TPUv3-512 | | | | | |
| 7B | 1 | 4 | 8 | **2048** | 256x |
| 13B | 1 | 2 | 8 | **1024** | 128x |
| TPUv4-1024 | | | | | |
| 3B | 8 | 16 | 32 | **16384** | 512x |
| 7B | 4 | 8 | 16 | **8192** | 512x |
| 13B | 4 | 8 | 16 | **4096** | 256x |
| 30B | 2 | 4 | 8 | **2048** | 256x |
| TPUv5e-256 | | | | | |
| 3B | 4 | 8 | 32 | **4096** | 128x |
| 7B | 2 | 8 | 16 | **2048** | 128x |

to prior state-of-the-arts, it is evident that we can train very large context models without compromising MFU or throughput.

## 5.3 Impact on In Context RL Performance

We present results of applying RingAttention for learning trial-and-error RL experience using Transformers. We report our results in Table 5, where we evaluate our proposed model on the ExoRL benchmark across six different tasks. On ExoRL, we report the cumulative return, as per ExoRL (Yarats et al., 2022). We compare BC, DT (Chen et al., 2021), AT (Liu and Abbeel, 2023a), and AT with memory efficient attention (Rabe and Staats, 2021) (AT+ME), AT with blockwise parallel transformers (Liu and Abbeel, 2023b) (AT+BPT), and AT with our RingAttention (AT+RingAttention). The numbers of BC, DT, AT are from the ExoRL and AT paper. AT + RingAttention numbers are run by ourselves. Since the ExoRL data is highly diverse, having been collected using unsupervised RL (Laskin et al., 2021), it has been found that TD learning performs best, while behavior cloning struggles (Yarats et al., 2022). AT (Liu and Abbeel, 2023a) shows that conditioning Transformer on multiple trajectories with relabeled target return can achieve competitive results with TD learning. For more details, please refer to their papers. We are interested in applying RingAttention to improve the performance of AT by conditioning on a larger number of trajectories rather than 32 trajectories in prior works. It is worth noting that each trajectory has $1000 \times 4$ length where 1000 is sequence length while 4 is return-state-action-reward, making training 128 trajectories with modest 350M size model infeasible for prior state-of-the-art blockwise parallel transformers. Results in Table 5 show that, by scaling up the sequence length (number of trajectories), AT + RingAttention consistently outperforms oringal AT with

**Table 4** Model flops utilization (MFU) with different training configurations: model sizes, compute, and context lengths. RingAttention enables training large models (7B-65B) on large input context sizes (over 4M) with negligible overheads.

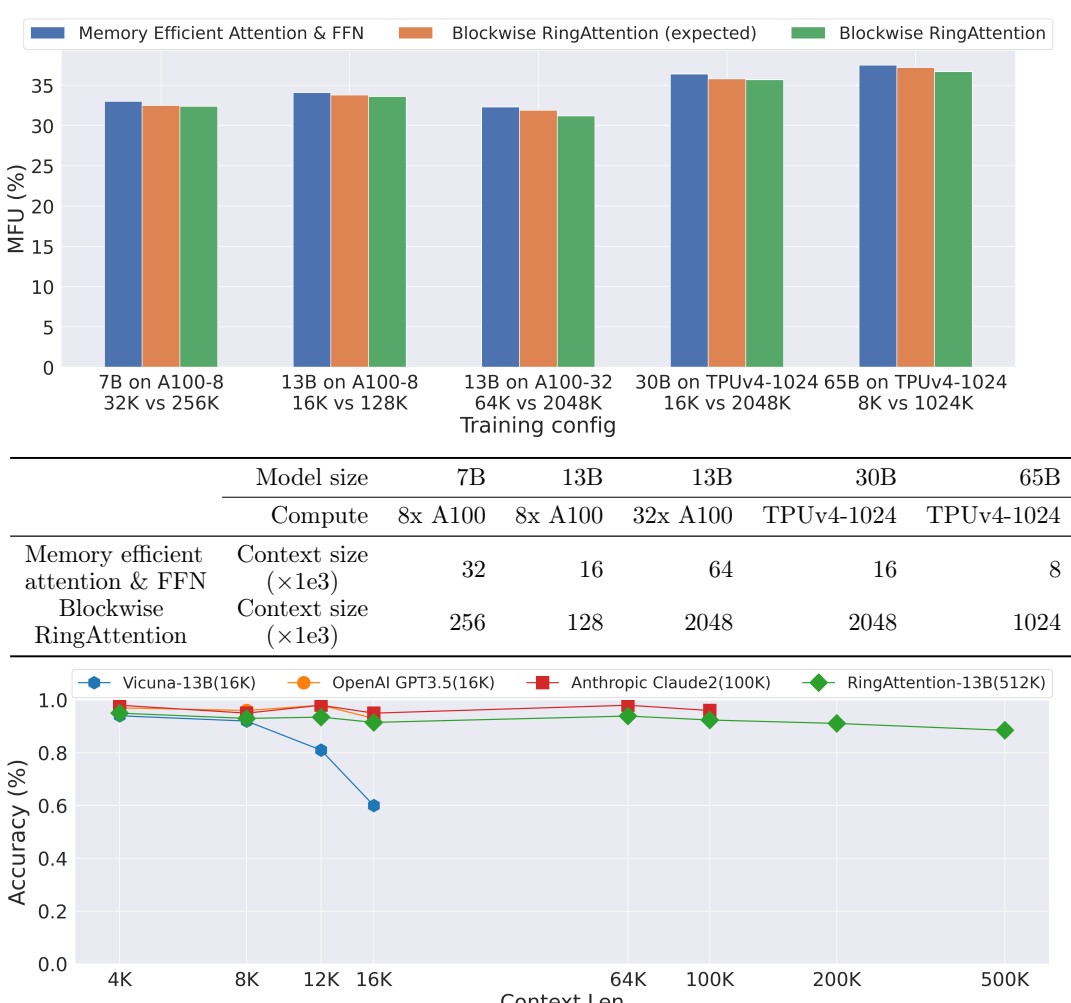

| | Model size | 7B | 13B | 13B | 30B | 65B |
|---|---|---|---|---|---|---|
| | Compute | 8x A100 | 8x A100 | 32x A100 | TPUv4-1024 | TPUv4-1024 |
| Memory efficient attention & FFN | Context size (×1e3) | 32 | 16 | 64 | 16 | 8 |
| Blockwise RingAttention | Context size (×1e3) | 256 | 128 | 2048 | 2048 | 1024 |

**Figure 3** Comparison of different models on the long-range line retrieval task.

BPT across all six tasks, achieving a total average return of 113.66 compared to the AT with BPT model's total average return of 111.13. The results show that the advantage of RingAttention for training and inference with long sequences.

**Table 5** Application of RingAttention on improving Transformer in RL. BC and DT use vanilla attention. AT + ME denotes using memory efficient attention, AT + BPT denotes using blockwise parallel transformer. AT + RA denotes using RingAttention.

| ExoRL | BC-10% | DT | AT + ME | AT + BPT | AT + BPT | AT + RA |
|---|---|---|---|---|---|---|
| **Task** | | | N Trajs = 32 | N Trajs = 32 | N Trajs = 128 | N Trajs = 128 |
| Walker Stand | 52.91 | 34.54 | oom | 95.45 | oom | 98.23 |
| Walker Run | 34.81 | 49.82 | oom | 105.88 | oom | 110.45 |
| Walker Walk | 13.53 | 34.94 | oom | 78.56 | oom | 78.95 |
| Cheetah Run | 34.66 | 67.53 | oom | 178.75 | oom | 181.34 |
| Jaco Reach | 23.95 | 18.64 | oom | 87.56 | oom | 89.51 |
| Cartpole Swingup | 56.82 | 67.56 | oom | 120.56 | oom | 123.45 |
| **Total Average** | 36.11 | 45.51 | oom | 111.13 | oom | **113.66** |

## 5.4 Impact on LLM Performance

We evaluate RingAttention by applying our method to finetune LLaMA model to longer context. In this experiment, while our approach enables training with millions of context tokens, we conducted finetuning on the LLaMA-13B model, limiting the context length to 512K tokens due to constraints on our cloud compute budget. This finetuning was carried out on 32 A100 GPUs, using the ShareGPT dataset, following methodologies as outlined in prior works (Chiang et al., 2023; Geng et al., 2023). We then evaluated our finetuned model on the line retrieval test (Li et al., 2023a). In this test, the model needs to precisely retrieve a number from a long document, the task can effectively capture the abilities of text generation, retrieval, and information association at long context, reflected by the retrieving accuracy. Figure 3 presents the accuracy results for different models across varying context lengths (measured in tokens). Notably, our model, RingAttention-13B-512K, stands out as it maintains high accuracy levels even with long contexts. GPT3.5-turbo-16K, Vicuna-16B-16K, and Claude-2-100K demonstrate competitive accuracy within short context lengths. However, they cannot handle extended context lengths.

## 6 Related Work

Transformers have garnered significant attention in the field of AI and have become the backbone for numerous state-of-the-art models. Several works have explored memory-efficient techniques to address the memory limitations of Transformers and enable their application to a wider range of problems. Computing exact self-attention in a blockwise manner using the tiling technique (Milakov and Gimelshein, 2018) has led to the development of memory efficient attention mechanisms (Rabe and Staats, 2021) and its efficient CUDA implementation (Dao et al., 2022), and blockwise parallel transformer (Liu and Abbeel, 2023b) that proposes computing both feedforward and self-attention block-by-block, resulting in a significant reduction in memory requirements. In line with these advancements, our work falls into the category of memory efficient computation for Transformers. Other works have investigated the approximation of attention mechanisms, yet these efforts have often yielded sub-optimal results or encountered challenges during scaling up. For an in-depth review of these techniques, we recommend referring to the surveys (Narang et al., 2021; Tay et al., 2022). Another avenue of research explores various parallelism methods, including data parallelism (Dean et al., 2012), tensor parallelism (Shoeybi et al., 2019), pipeline parallelism (Narayanan et al., 2019; Huang et al., 2019; Narayanan et al., 2021), sequence parallelism (Li et al., 2023b; Korthikanti et al., 2022; Jacobs et al., 2023), and FSDP (Facebook, 2023; Rajbhandari et al., 2020). The activations of self-attention take a substantial amount of memory for large context models. Tensor parallelism can only reduce parts of activations memory and sequence parallelism introduces a significant communication overhead that cannot be fully overlapped with computation. Prior work has studied sharding along sequence and attention heads, and gathering sequences via an optimized all-to-all topology, achieving reduced communication (Jacobs et al., 2023). However, this method is restricted by the number of attention heads and requires gathering the full sequence on each device. In comparison, our approach fully overlaps communication with blockwise computation, enhancing its scalability. Prior work study sequence parallelism for computing self-attention using a ring topology (Li et al., 2023b), but is not optimized for blockwise parallel transformers and is incompatible with memory-efficient attention, which are crucial to large context training. Overlapping communication with computation remains challenging, and the communication overheads make it infeasible for training and inference in large-context scenarios. Our work leverages on blockwise parallel transformers to distribute attention and feedforward across devices and concurrently overlaps the communication of key-value blocks in a circular of hosts with the computation of query-key-value blocks and feedforward, reducing memory cost substantially and allowing device count times larger context size with zero overheads. Overlapping communication with computation has been studied in high performance computing literature (Danalis et al., 2005; Wang et al., 2022; Danalis et al., 2009, *inter alia*). While ring communication has found applications in other parallel computing scenarios (Bischof, 2008; Hursey and Graham, 2011; Gibiansky, 2017; Sergeev and Del Balso, 2018), our work stands out as the first work to show that it can be applied to

self-attention as used in Transformers and to make it fit efficiently into Transformer training and inference without adding significant overhead by overlapping blockwise computation and communication.

## 7 Conclusion

In conclusion, we propose a memory efficient approach to reduce the memory requirements of Transformers, the backbone of state-of-the-art AI models. Our approach allows the context length to scale linearly with the number of devices while maintaining performance, eliminating the memory bottleneck imposed by individual devices. Through extensive experiments on language modeling and reinforcement learning, we demonstrate its effectiveness, enabling training sequences that are up to device count times longer than those of prior memory-efficient Transformers, exceeding a context length of 100 million without making approximations to attention. In terms of future prospects, the possibility of near-infinite context introduces a vast array of exciting opportunities, such as large video-audio-language models, learning from extended feedback and trial-and-errors, understanding and generating codebase, adapting AI models to understand scientific data such as gene sequences, and developing strong reasoning from link gathering data.

## Acknowledgments

This project is supported in part by Office of Naval Research grant N00014-21-1-2769. We express our gratitude to the BAIR and RLL communities for their insightful discussions and feedback. We are also thankful to David Patterson for addressing our questions about TPUs and giving insightful feedback on early versions of this work. Our appreciation goes out to Yash Katariya and Sharad Vikram from the Jax developers' team for assisting with our Jax related questions. We also thank Tri Dao for the valuable feedback on this work. We thank Google TPU Research Cloud for granting us access to TPUs.

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

# A  CODE

The implementation of RingAttention in Jax is provided in Figure 4. We use `defvjp` function to define both the forward and backward passes, and use collective operation `jax.lax.ppermute` to facilitate the exchange of key-value blocks among a ring of hosts. The provided code snippet highlights essential components of RingAttention. We provide the complete code on github.

For large scale end-to-end training on TPU or on GPU cluster with high bandwidth inter connection, we recommend using FSDP to shard large models and using RingAttention to achieve large context. If total batch size is too large, add tensor parallelism to reduce the global batch size. The degree of parallelism can be adjusted using the `mesh_dim` parameter within the codebase. To illustrate, consider a setup with 512 devices, such as 512x A100. If the model size is 30B, you can shard it across 8 devices and allocate the remaining 32 devices for RingAttention. This setup allows the context size to be expanded 32 times more than if you didn't use RingAttention. Conversely, for models sized 7B or 3B, there is no need for FSDP. This means you can utilize all 512 devices exclusively to expand the context using RingAttention by 512 times. Building upon the result that our approach allows for a 256K context size when using 8x A100 GPUs, it suggests that by employing 512 A100 GPUs, the potential context size can be expanded to 16 million.

# B  EXPERIMENT DETAILS

## B.1  EVALUATION OF CONTEXT LENGTH

In the experimental results presented in Section 5.1, we used fully sharded tensor parallelism (FSDP) to partition the model across GPUs or TPU devices. Our evaluation focused on determining the maximum achievable sequence length in commonly used FSDP training scenarios. For TPUs, we utilized its default training configuration, which involved performing matmul operations in `bfloat16` format with weight accumulation in `float32`. On the other hand, for GPUs, we adopted the default setup, where all operations were performed in `float32`.

## B.2  EVALUATION OF MFU

In the evaluation presented in Section 5.2. The batch size in tokens is 2 million per batch on GPU and 4 million per batch on TPU. The training was conducted using FSDP Facebook (2023) with Jax SPMD. For gradient checkpointing (Chen et al., 2016), we used `nothing_saveable` as checkpointing policies for attention and feedforward network (FFN). For more details, please refer to Jax documentation.

## B.3  EVALUATION ON LINE RETRIEVAL

In the evaluation presented in Section 5.4, we finetuned the LLaMA-13B model (Touvron et al., 2023), limiting context length to 512K tokens due to constraints on our cloud compute budget, the training was conducted on 32x A100 80GB Cloud GPUs. We use user-shared conversations gathered from ShareGPT.com with its public APIs for finetuning, following methodologies as outlined in prior works (Chiang et al., 2023; Geng et al., 2023). ShareGPT is a website where users can share their ChatGPT conversations. To ensure data quality, we convert the HTML back to markdown and filter out some inappropriate or low-quality samples, which results in 125K conversations after data cleaning.

# C  INFERENCE REQUIREMENT

We provide the minimal sequence length required to overlap communication with computation during training in Table 2. Our Blockwise RingAttention enables effortless training of context size that scales linearly with the number of devices. While we focus on introducing training as it is more memory demanding than autoregressive inference where the number of query

```python
1   def _ring_attention_fwd(q, k, v, attn_bias, axis_name, float32_logits, blockwise_kwargs):
2       if float32_logits:
3           q, k = q.astype(jnp.float32), k.astype(jnp.float32)
4       batch, q_len, num_heads, dim_per_head = q.shape
5       batch, kv_len, num_heads, dim_per_head = k.shape
6       numerator = jnp.zeros((batch, q_len, num_heads, dim_per_head)).astype(q.dtype)
7       denominator = jnp.zeros((batch, num_heads, q_len)).astype(q.dtype)
8       axis_size = lax.psum(1, axis_name)
9       block_size = q_len # assumes this function is pre-sharded inside shard_map
10      query_chunk_size = blockwise_kwargs["query_chunk_size"]
11      key_chunk_size = blockwise_kwargs["key_chunk_size"]
12      def scan_kv_block(carry, idx):
13          prev_max_score, numerator, denominator, k, v = carry
14          attn_bias_slice = lax.dynamic_slice_in_dim(attn_bias,
15              (lax.axis_index(axis_name) - idx) % axis_size * kv_len, kv_len, axis=-1)
16          q_block_idx = lax.axis_index(axis_name)
17          k_block_idx = (lax.axis_index(axis_name) - idx) % axis_size
18          q_chunk_idx_start = q_block_idx * (block_size // query_chunk_size)
19          k_chunk_idx_start = k_block_idx * (block_size // key_chunk_size)
20          numerator, denominator, max_score = _blockwise_attention_fwd(q, k, v,
21              (numerator, denominator, prev_max_score), q_chunk_idx_start, k_chunk_idx_start,
22              bias=attn_bias_slice, **blockwise_kwargs)
23          k, v = map(lambda x: lax.ppermute(x, axis_name, perm=[(i, (i + 1) % axis_size)
24              for i in range(axis_size)]), (k, v))
25          return (max_score, numerator, denominator, k, v), None
26      prev_max_score = jnp.full((batch, num_heads, q_len), -jnp.inf).astype(q.dtype)
27      (max_score, numerator, denominator, _, _), _ = lax.scan(scan_kv_block,
28          init=(prev_max_score, numerator, denominator, k, v), xs=jnp.arange(0, axis_size))
29      output = numerator / rearrange(denominator, 'b h q -> b q h')[..., None]
30      return output.astype(v.dtype), (output, q, k, v, attn_bias, denominator, max_score)
31
32  def _ring_attention_bwd(axis_name, float32_logits, blockwise_kwargs, res, g):
33      output, q, k, v, attn_bias, denominator, max_score = res
34      batch, kv_len, num_heads, dim_per_head = k.shape
35      axis_size = lax.psum(1, axis_name)
36      dq = jnp.zeros_like(q, dtype=jnp.float32)
37      dk = jnp.zeros_like(k, dtype=jnp.float32)
38      dv = jnp.zeros_like(v, dtype=jnp.float32)
39      query_chunk_size = blockwise_kwargs["query_chunk_size"]
40      key_chunk_size = blockwise_kwargs["key_chunk_size"]
41      block_size = q.shape[1] # assumes this function is pre-sharded inside shard_map
42      def scan_kv_block(carry, idx):
43          dq, dk, dv, k, v = carry
44          attn_bias_slice = lax.dynamic_slice_in_dim(attn_bias,
45              (lax.axis_index(axis_name) - idx) % axis_size * kv_len, kv_len, axis=-1)
46          q_block_idx = lax.axis_index(axis_name)
47          k_block_idx = (lax.axis_index(axis_name) - idx) % axis_size
48          q_chunk_idx_start = q_block_idx * (block_size // query_chunk_size)
49          k_chunk_idx_start = k_block_idx * (block_size // key_chunk_size)
50          dq, dk, dv = _blockwise_attention_bwd(q, k, v, g, (dq, dk, dv, output, denominator, max_score),
51              q_chunk_idx_start, k_chunk_idx_start, bias=attn_bias_slice, **blockwise_kwargs)
52          k, v, dk, dv = map(lambda x: lax.ppermute(x, axis_name, perm=[(i,
53              (i + 1) % axis_size) for i in range(axis_size)]), (k, v, dk, dv))
54          return (dq, dk, dv, k, v), None
55      (dq, dk, dv, k, v), _ = lax.scan(scan_kv_block, init=(dq, dk, dv, k, v), xs=jnp.arange(0, axis_size))
56      dq, dk, dv = dq.astype(q.dtype), dk.astype(k.dtype), dv.astype(v.dtype)
57      return dq, dk, dv, None
58
59  @partial(jax.custom_vjp, nondiff_argnums=[4, 5, 6])
60  def ring_attention(q, k, v, attn_bias, axis_name, float32_logits, blockwise_kwargs):
61      y, _ = _ring_attention_fwd(q, k, v, attn_bias, axis_name, float32_logits, blockwise_kwargs)
62      return y
63
64  ring_attention.defvjp(_ring_attention_fwd, _ring_attention_bwd)
```

**Figure 4** Key parts of the implementation in Jax. We use collective operation `lax.ppermute` to send and receive key value blocks between previous and next hosts. The full code is implemented in Jax and Pallas for best performance.

token is one, RingAttention is applicable to inference too. For example, serving a LLaMa 7B on 32x TPUv5e, the conventional approach is to distribute the model along the attention heads dimension, with each device computing one attention head. Assuming a batch size of 1, this can serve up to a 256K context length due to key-value cache activation size. RingAttention can allow 32 times larger context by circulating the key-value cache between a ring of devices. To overlap the communication with computation, it needs d2/F >= 2*d2/B, where B/F >=2. With a bandwidth of 186 GB/s and flops of 196 TFLOPs, and assuming an unreasonably high MFU of 40% for this large context, then B/F = 2.4, meaning that RingAttention allows 32 times larger context for inference without adding overheads.

## D  Training FLOPs Scaling of Context Size

Given that our proposed approach unlocks the possibility of training with a context size exceeding 100 million tokens and allows for linear scaling of the context size based on the number of devices, it is essential to understand how the training FLOPs per dataset scale with the context size. While a larger context size results in a higher number of FLOPs, the increased ratio does not scale quadratically because the number of tokens remains fixed. We present these results in Figure 5, which showcases various model sizes and context lengths, representing different computational budgets. The figure shows the ratio of FLOPs for larger context lengths compared to the same model with a shorter 4K context size. We calculated the

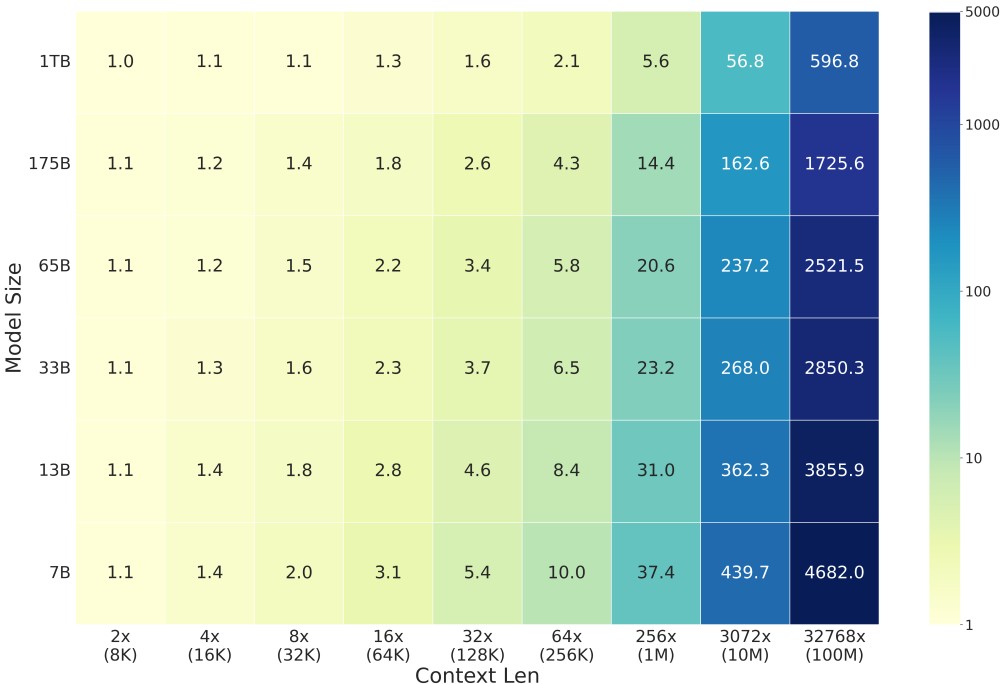

**Figure 5**  The per dataset trainig FLOPs cost ratio relative to a 4k context size, considering different model dimensions. On the x-axis, you'll find the context length, where, for example, 32x(128k) denotes a context length of 128k, 32x the size of the same model's 4k context length.

per sequence FLOPs using $(24bsh^2 + 4bs^2h)n$ where $h$ is model hidden dimension, $b$ is batch size, $s$ is total sequence length, and $n$ is number of layers. The per dataset FLOPs ratio is then given by $((24bs_2h^2 + 4bs_2{}^2h)/(24bs_1h^2 + 4bs_1{}^2h))/(s_2/s_1) = (6h+s_2)/(6h+s_1)$, where $s_2$ and $s_1$ are new and old context lengths. Model sizes and their hidden dimensions are as follows: LLaMA-7B (4096), LLaMA-13B (5140), LLaMA-33B (7168), LLaMA-65B (8192), GPT3-175B (12288), and 1TB (36864). These model configurations are from LLaMA (Touvron et al., 2023) and GPT-3 (Brown et al., 2020) papers, except the 1TB model size and dimension were defined by us.

As depicted in Figure 5, scaling up small models to a 1M context size results in approximately 20-40 times more FLOPs, and even more for 10M and 100M token context sizes. However, as the model sizes increase, the cost ratio decreases. For instance, scaling up the 170B model from 4K to 10M incurs 162.6x higher per dataset FLOPs, despite the context size being 3072 times longer.

