# OpenReview forum: "RingAttention with Blockwise Transformers for Near-Infinite Context"
_ICLR.cc/2024/Conference — ICLR 2024 poster_

### Official Review · Reviewer_Z7SR · 2023-10-30

**Soundness:** 2 fair
**Presentation:** 3 good
**Contribution:** 3 good
**Rating:** 5
**Confidence:** 4

**Summary:**

Using large context lengths poses memory challenges since it is needed to maintain the key value vectors of earlier tokens. This paper proposes to distribute these vectors across multiple nodes allowing linearly scaling the context length with number of nodes. To address the challenge that computing the attention for a token requires access to all previous key value vectors, the authors propose a scheme to move the vectors across the node. Namely, they suggest forming a ring of nodes and passing the vector to the next node in the ring continuously until all vectors pass through all nodes. Moreover, the authors point out that the communication can be overlapped with the computation, thus making the overhead negligible. The paper shows the method can be practically applied through various experiments, namely fine-tuning LLaMA with 500k token context length.

**Strengths:**

The method effectively allows scaling the context length with the number of nodes. The proposal to overlap communication and computation could allow hiding the communication cost. It can also be implemented in a fairly straightforward manner. The authors show Ring Attention's practicality through experiments. It is very nice to see experiments done both on GPUs and TPUs. The paper is written clearly and can be followed reasonably easily.

**Weaknesses:**

While the paper contains experiments showcasing that ring attention can be applied in practice, a comparison with other methods is missing. For example, if applicable, is it better to apply pipeline parallelism or ring attention? Other similar methods also include DeepSpeed Inference [1] which suggests to offload the kv cache to cpu and similarly overlap communication and computation and FlexGen [2]. Additionally, in light of these existing work (some of which are also not referenced), the novelty of the work might be limited.

Furthermore, an ablation study is missing. For example, how important is it to overlap communication and computation?

I would also like to point out that while the method allows scaling the context length by increasing the number of nodes, it does not address challenges of handling long contexts by Transformers (e.g. [3]). As such, I find comparison with Claude and GPT3.5 in Figure 3 a bit misleading. In particular, we do not expect this method to improve the performance of the model on small context lengths (though a small increase is observed possibly due to more fine-tuning).

[1] Aminabadi, Reza Yazdani, Samyam Rajbhandari, Ammar Ahmad Awan, Cheng Li, Du Li, Elton Zheng, Olatunji Ruwase et al. "DeepSpeed-inference: enabling efficient inference of transformer models at unprecedented scale." In SC22: International Conference for High Performance Computing, Networking, Storage and Analysis, pp. 1-15. IEEE, 2022.

[2] Sheng, Ying, Lianmin Zheng, Binhang Yuan, Zhuohan Li, Max Ryabinin, Beidi Chen, Percy Liang, Christopher Re, Ion Stoica, and Ce Zhang. "FlexGen: High-Throughput Generative Inference of Large Language Models with a Single GPU." (2023).

[3] Liu, Nelson F., Kevin Lin, John Hewitt, Ashwin Paranjape, Michele Bevilacqua, Fabio Petroni, and Percy Liang. "Lost in the middle: How language models use long contexts." arXiv preprint arXiv:2307.03172 (2023).

Some minor suggestions:
* Table 1 is not table but a figure.
* The last paragraph of section 2 is an almost replica of the introduction (this is not a problem on its own. I am merely pointing this out in case the authors would like to make better use of the space)

**Questions:**

(Please also see the weakness section.)

1. How is ring attention applied while doing auto regressive decoding (in the stage of decoding tokens one by one)?

2. How does the training time scale with number of nodes? For example, what is the rate of slowdown observed when training a model with 2k context length on two nodes in comparison with k context length on one node (expected 4x) and in comparison with using normal attention to train with 2k context length on a single node?

---

> ### Author Response · Authors · 2023-11-20
> **Response to reviewer Z7SR**
>
> Dear reviewer Z7SR,
>
> Thank you so much for your review and for highlighting the effectiveness of our method, the comprehensiveness of our experiments, and the clear, easily-followable writing of our paper. We appreciate your feedback and suggestions, which we have incorporated into the revised version to improve it.
>
>
> **#1: Relation to DeepSpeed-inference and FlexGen**
>
> There are some key differences:
>
> 1. CPU memory and the CPU-GPU interconnect are much slower than the GPU-GPU interconnect Ring Attention utilizes. Thus, using CPU offloading would not work for large context training and inference.
>
> 2. Ring Attention allows training and inference with a context size that scales linearly with device count, while prior CPU offloading is only applicable to inference. Ring Attention does not add communication overheads while prior CPU offloading addes large communication costs.
>
> 3. DeepSpeed-inference and FlexGen require storing the entire full sequence on each device, while Ring Attention distributes a long sequence to a ring of devices, and never needs to store full sequence on any device. This allows Ring Attention to process device count times longer sequences.
>
> 4. Due to limitations in physical space and CPU memory, offloading to the CPU is insufficient for managing long sequence lengths. The small CPU-accelerator bandwidth necessitates more stringent requirements for hiding communication in CPU offloading, reducing its flexibility and applicability. In contrast, Ring Attention, which is widely applicable on GPU and TPU, requires only a fast interconnect between nearby devices, and keeps all data fast memory only while CPU offloading saves data to slower CPU memory. This minimal requirement enables Ring Attention to be scalable to any number of devices. Unlike CPU offloading that necessitates storing the full sequence on each device, Ring Attention distributes the sequence across multiple devices.
>
>
> **#2: While the paper contains experiments showcasing that ring attention can be applied in practice, a comparison with other methods is missing. For example, if applicable, is it better to apply pipeline parallelism or ring attention?**
>
> Other parallelism methods, like data, pipeline, tensor parallelism, and FSDP, necessitate storing entire sequences on each device, rendering them incapable of processing large context sizes. Pipeline parallelism also leads to device idle time due to non-overlapped communication. Sequence parallelism distributes the sequence across devices but requires extensive communication that cannot be overlapped, making it infeasible for large context sizes.
>
> We have also included a comparison with Deep Speed Ulysses, which was published on Arxiv three days prior to the ICLR submission deadline. Deep Speed Ulysses combines sequence parallelism and tensor parallelism for their optimized NVSwitch all-to-all topology. It can reduce the communication cost of sequence parallelism. Importantly, it requires gathering and storing full sequence on each device, while Ring Attention distributes long sequences to a ring of devices and never needs to keep full sequence on any device. Ring Attention outperforms it substantially both in maximum context size and training efficiency. Please refer to our response to reviewer haGA for the full results.
>
> Ring Attention is the first work that distributes transformers computation on long sequences across devices and enables fully overlapped communication with computation. Ring Attention is orthogonal and fully compatible with existing large-scale training parallelism methods.
>
> We demonstrate this by applying Ring Attention to FSDP end-to-end large-scale training of 7B-65B LLM on 32-1024 TPUv4, covering various compute budgets. Ring Attention facilitates the effortless training of very large context sizes without incurring overheads or making approximations. For example, on 1024 TPUv4, Ring Attention enables training with context sizes over millions in tokens, which is 512 times larger than prior state-of-the-art memory efficient / flash attention, without adding communication nor computation overheads.

---

> > ### Author Response · Authors · 2023-11-20
> > **Response to reviewer Z7SR (continued)**
> >
> > **#3: How is ring attention applied while doing auto regressive decoding (in the stage of decoding tokens one by one)?**
> >
> > During inference, using tensor parallelism on the attention heads' dimension allows distribution of LLM across devices up to the number of attention heads, with each device computing a subset of these heads. Consider the 7B LLaMA, which has 32 attention heads; this should allow 256K context length inference on 32x TPUv5e. By adding Ring Attention to distribute the kv cache between devices, it becomes clear that the maximum context length can be increased by 32 times, reaching 8M tokens in length. To overlap the communication with computation, we need d2/F >= 2*d2/B, where B/F >=2, the bandwidth is 186 GB/s and flops are 196 TFLOPs. Assuming a very high MFU of 40% for this large context, then B/F = 2.4, indicating that Ring Attention allows a 32 times larger context for inference without adding overheads. We will clarify this further in the revision and thank the reviewer for the question.
> >
> >
> > **#4: Long context impact on model performance**
> >
> > The model trained with Ring Attention maintains competitive accuracy levels comparable to Claude and GPT4 and can process sequences 5-16 times longer. We include results of applying Ring Attention to extend the LLaMA context length to 512K tokens and evaluate its performance using LongChat tasks, as shown in Figure 3. Following the reviewer’s request, we also studied model performance on additional tasks, including training agentic transformers with more trajectories for improved in-context RL. The results in Table 5 demonstrate that a large context window significantly enhances the performance of learning decision-making policies from large-scale experiences. A large context allows agentic transformers to encode many more trajectories simultaneously, making hindsight relabeling more informative and enabling transformers to learn policies more effectively.
> >
> > Regarding the “lost in the middle” observation, it is an interesting observation that some LLMs may not fully leverage the full context information. This observation is orthogonal to our paper, as our work focuses on making it possible to train long-context transformers without making approximations. Ring Attention could enable academic and industry labs to train long-context transformers to further investigate this phenomenon. Anthropic’s blog (https://www.anthropic.com/index/prompting-long-context#b3) suggests a different trend compared to the “lost in the middle” observation. This discrepancy might relate to task-specific characteristics, where additional texts tend to contain relevant information primarily at the beginning or end due to human biases, indicating a potential data issue. These are intriguing research areas orthogonal to Ring Attention, and Ring Attention can facilitate future research on the impact of long contexts on model performance.
> >
> >
> > Please let us know if our response resolves your concerns. We look forward to hearing from you. Thank you so much.

---

### Official Review · Reviewer_rUcK · 2023-10-30

**Soundness:** 3 good
**Presentation:** 3 good
**Contribution:** 3 good
**Rating:** 8
**Confidence:** 3

**Summary:**

The paper proposed a distributed computing strategy for distributing self-attention computation to across multiple devices without introducing latency on host communication. The authors observed that when distributing blockwise computation of self-attention in multiple devices, the order of which block is computed first does not change the output. So, they proposed a ring style communication. Each device will take care of self-attention output of a subset of queries, and it only receives a subset of keys and values from the next device at each time, which reduces the communication cost. Also, by overlapping the computation of queries’ attention to the current subset of keys and values and the communication of the next subset of keys and values, this host communication does not introduce additional latency overhead.

**Strengths:**

1. The idea is simple and effective.

2. The proposed distributed self-attention computation allows bypassing the hardware limitation of single device.

3. The ring style communication allows the communication requirement stays constant when scaling to more devices.

4. The overlapping of computation and communication hides the communication overhead.

**Weaknesses:**

1. There is only one toy experiment comparing the performance of RingAttention with other efficient attention baselines. The proposed method enables full self-attention for longer sequences, so it would be interesting to see the performance potential of full self-attention on the real world datasets.

2. It would be better to include the results on network bandwidth utilizations in Table 3.

3. In causal attention (used in most LLMs), the compute cost for different queries are different. Queries at the begin of sequence require less compute. In the proposed method, since each device takes care of a subset of queries, the compute load on different devices are different. Will this method make some devices underutilized?

**Questions:**

1. I am wondering if the same idea could be adopted to distribute linear projections in Attn and FFN computations as well

---

> ### Author Response · Authors · 2023-11-20
> **Response to reviewer rUcK**
>
> Dear Reviewer rUcK,
>
>
> Thank you so much for the positive review and for highlighting that our work is simple and effective. We appreciate your feedback and suggestions, which we have incorporated into the revised version to improve it.
>
>
> **#1: The proposed method enables full self-attention for longer sequences, so it would be interesting to see the performance potential of full self-attention on the real world datasets.**
>
> Thank you for the suggestion. We have conducted additional experiments on more tasks. We applied Ring Attention to train agentic transformers on hundreds of trajectories to achieve better in-context RL. The results in Table 5 show that a large context window significantly improves performance of learning decision making policy from large scale experience. Large context allows agentic transformers to encode many more trajectories at the same time, so the hindsight relabeling is more informative and makes transformers learn a policy more effectively.
>
>
> **#2: It would be better to include the results on network bandwidth utilizations in Table 3.**
>
> Thanks for suggesting to report network utilization rate in addition to maximum context sizes in end-to-end large-scale training, which can provide more information about the training.
> We have included the measurement of network utilization rates in Table 5. We measure the bandwidth utilization using \texttt{jax.profiler} and, based on the average measurements from 3 runs. The results show that Ring Attention has significantly higher bandwidth utilization rates. This is attributed to more communication from larger context sizes.
>
>
> **#3: Balancing compute load on different devices for causal attention (used in most LLMs)**
>
> You are right that in causal attention mode, our method will make some devices underutilized. This is a great, insightful suggestion to Ring Attention. By skipping upper triangular blocks computation and balancing the computation load between devices when using causal attention, the compute cost can be reduced and potentially leading to a twofold increase in speed. We are definitely interested in this and planning to research this direction in future work.
>
>
> **#4: Whether the same idea could be adopted to distribute other computations as well.**
>
> We focus on self-attention in this work, since it contributes the largest activation size in today’s state-of-the-art AI. We believe the idea of Ring Attention can be applied to any all-to-all ops and even mixture of experts or split up a FFN computation or some aggregation.
>
>
> Please let us know if our response resolves your concerns. We look forward to hearing from you. Thank you so much.

---

### Official Review · Reviewer_dMuy · 2023-10-31

**Soundness:** 3 good
**Presentation:** 3 good
**Contribution:** 1 poor
**Rating:** 3
**Confidence:** 5

**Summary:**

The paper proposes ring attention, which is a way of implementing softmax dense attention.  The attention operation is
distributed over multiple GPUs, and computed in parallel along the sequence dimension.  Attention is also computed in a blockwise fashion, without ever instantiating the full attention matrix, which reduces memory requirements. Together, these two techniques allow dense attention to be scaled to very long sequences.

**Strengths:**

The paper is well-written, and it cites the appropriate literature.  The technique described in this paper does, in fact, allow transformers to be trained with very long sequences.

**Weaknesses:**

Unfortunately, although the paper is technically sound, it does not really contribute anything new.  The ideas presented
in this paper will already be obvious to most practitioners.  As a result, I do not feel that it is appropriate for publication at a conference.

The fact that transformers process all elements of a sequence in parallel, (at least during training) was the whole inspiration for the original 2017 paper that introduced the transformer architecture.  Thus, the idea of distributing the computation in parallel along sequence length is most definitely nothing new.  This paper merely distributes the computation over multiple devices.

A naive implementation of attention instantiates the entire attention matrix, which is too large to fit in memory for long sequences.  The blockwise computation of dense attention avoids such instantiation, and it does involve some subtle implementation details wrt. to the handling of softmax.  However, it is also not a contribution of this paper; the authors cite the appropriate prior work.

The remaining piece of the puzzle is to overlap the matrix multiplications for each block with the communication overhead of transferring blocks between different hosts.  However, this is also nothing new; tiled matrix multiplication, distributed over multiple GPUs, is implemented by every major machine learning library today (e.g. Jax), and existing implementations overlap matmuls and communication.

Thus, the authors merely observe that it is possible to implement long-range attention using a simple combination of well-understood and previously published techniques.  This fact should be obvious to most practitioners, and IMO, does not rise to the level of a conference paper.  I would encourage the authors to publish this work in a workshop or on arXiv.

**Questions:**

The authors claim that ring attention over very long sequences is suitable for both training and inference.  However, the cost of attention in both cases is still O(N^2).  For training, this is less of an issue; all elements of the (long) sequence are processed in parallel, so the sequence length is essentially just a batch dimension, and the number of tokens per training step is not excessive for an LLM.

However, for inference, tokens are processed autoregressively, so very long sequence lengths will lead to very slow generation times, and thus very high latency when serving the model.  The authors do not discuss or even mention this problem.  How does the latency of ring attention over, e.g., 8M tokens during inference compare to a more conventional short-context model?

---

> ### Author Response · Authors · 2023-11-20
> **Response to reviewer dMuy**
>
> Dear reviewer dMuy,
>
> Thank you so much for the review. We believe that there are misunderstandings. We appreciate your feedback and suggestions, which we have incorporated into the revised version to improve it.
>
>
> **#1: This is also nothing new; tiled matrix multiplication, distributed over multiple GPUs, is implemented by every major machine learning library today (e.g. Jax), and existing implementations overlap matmuls and communication.**
>
> We respectfully disagree with the reviewer’s criticism regarding novelty. To the best of our knowledge, Ring Attention is the first work that proposes distributing blockwise transformers in a ring topology and overlaps communication with computation. It is also the first to enable training transformers on extremely long sequences without incurring extra computation or communication costs. Ring Attention allows the context size to scale linearly with the device count, addressing the longstanding challenge of freeing transformers from memory and context limitations. For example, Ring Attention enables training LLaMa with a context size of more than millions of tokens on 1,024 TPUv4, which is 512 times larger in context size compared to the previous state-of-the-art blockwise transformers and memory-efficient/flash attention. In contrast, today’s state-of-the-art commercial models have much more limited context sizes, such as GPT-4-turbo has only 128K.
>
> We believe that an effective, novel, and intuitive idea should be encouraged rather than discouraged.
>
> We would like to point out that attention is not tiled matrix multiplication since it involves softmax, that is why the ML community has published memory efficient attention which computes attention cumulatively to allow long sequence and flash attention which optimizes the speed while allowing long sequence. However, these works require storing full sequences on each device, making it impossible for processing long sequences such as books, videos, and trajectories, among others.
>
> Past published work in the ML community has suggested less scalable solutions, such as sequence parallelism, tensor parallelism, and fully sharded data parallelism. These methods are less scalable because they require storing full sequences on each device (as in tensor parallelism and fully sharded data parallelism) or incur large communication costs that cannot be overlapped (as in sequence parallelism). Ring Attention, however, is an innovative approach that distributes long sequences across a ring of devices and never needs to store the full sequence on any single device. It is the first work that enables linear scaling up of context size with the number of devices for extremely long sequences without incurring extra computation or communication costs.
>
> A second important contribution of our paper is the first set of experiments with transformers trained to this massive context length, showing good performance on several ML tasks. We demonstrate that Ring Attention enables us to train Vicuna chatbot to 512K context size with just 32 GPUs, while prior state-of-the-art memory-efficient/flash attention-based methods can only handle 16K context size. This is a 32-fold increase in context window size compared to the previous long context Vicuna. The model's performance is competitive with GPT-4 and Claude-2 in long-context line retrieval tasks, and it can process sequences over five times longer than those models. We also applied Ring Attention to scale agentic transformers across hundreds of hindsight-relabeled trajectories. This approach led to substantially improved in-context RL performance, achieving state-of-the-art results in in-context RL tasks.
>
> Apart from proposing the idea of Ring Attention, this work provides an open-source implementation, which is plug-and-play and easy to use.
>
> In sum, Ring Attention is the first work proposing the distribution of blockwise transformers in a ring topology and overlaps communication with computation. This approach addresses the long-standing memory challenge of transformers without making approximations, allowing the context size to scale linearly with device count without additional computation or communication costs.
>
>
> **#2: The authors claim that ring attention over very long sequences is suitable for both training and inference. However, the cost of attention in both cases is still O(N^2).**
>
> Since we do not make approximations to attention, the compute cost is indeed still O(N^2). Our aim is to make it possible to fit long sequences into transformers, i.e., addressing the memory challenge. Combining Ring Attention with approaches that reduce compute costs, such as sparse attention, is a promising direction, but our work makes even precise attention efficiently scalable to sequence lengths in the millions.

---

> > ### Author Response · Authors · 2023-11-20
> > **Response to reviewer dMuy (continued)**
> >
> > **#3: How does the latency of ring attention over, e.g., 8M tokens during inference compare to a more conventional short-context model?**
> >
> > Ring Attention does not add latency overheads. For example, serving a LLaMa 7B on 32x TPUv5e, the conventional approach is to distribute the model along the attention heads dimension, with each device computing one attention head. Assuming a batch size of 1, this can serve up to a 256K context length due to kv cache activation size. Ring Attention can allow 32 times larger context by circulating the kv cache between a ring of devices. To overlap the communication with computation, it needs d2/F >= 2*d2/B which requires B/F >=2. With a bandwidth of 186 GB/s and flops of 196 TFLOPs, and assuming an unreasonably high MFU of 40% for this large context, then B/F = 2.4 which is >2, meaning that Ring Attention allows 32 times larger context for inference without adding overheads. Ring Attention can also be combined with KV caching techniques such as vLLM for efficient inference on repeated prefixes.
> >
> > In terms of latency in specific applications, 8M tokens inference is slower than short-context inference due to the increased computation required for long-context. For applications requiring long sequences, such as processing videos and agents' trajectories, the latency will indeed be higher than processing simple chitchat queries. We believe that the latency is independent of our work, as Ring Attention focuses on making large context training and inference possible, a feat previous methods could not achieve.
> >
> > We are definitely interested in exploring ways to reduce compute costs and therefore latency in future research, such as by leveraging sparsity in transformers.
> >
> >
> > Please let us know if our response resolves your concerns. We look forward to hearing from you. Thank you so much.

---

> > ### Comment · Reviewer_dMuy · 2023-11-20
> >
> > Thank you for your response.  Unfortunately, I must stand by my original assessment.  Sharding along sequence length is not particularly different from sharding along any other dimension wrt. to doing a matrix multiply.   As I mentioned in my original review, dealing with softmax does require a mathematical trick to avoid instantiating the full attention matrix, but that trick is not a contribution of this paper.  The ring topology of TPUs is used in the same way by existing parallel matrix multiplication libraries.
> >
> > BTW, I do not mean to disparage the software engineering effort required to implement Ring Attention in any way.   If you were to release it as an open-source library, I believe it would be a valuable contribution to the industry.  I also appreciate your observation that there is not necessarily any need to switch to fancy new attention mechanisms, when conventional attention can be implemented at long range, simply by sharding the computation along a different dimension.

---

> > > ### Author Response · Authors · 2023-11-20
> > > **Response to reviewer dMuy**
> > >
> > > Dear reviewer dMuy,
> > >
> > > Thank you for your reply. We regret that our response did not address your concerns. It’s disappointing that, based on the opinions, many useful techniques in the ML community would be dismissed as “nothing new”, such as: memory efficient attention [1], which would merely be the already known online softmax [2], and flash attention [3], which would be simply a CUDA implementation of the previously mentioned memory efficient attention [1].
> > >
> > > [1] Self-attention Does Not Need O(n2) Memory. https://arxiv.org/abs/2112.05682
> > >
> > > [2] Online normalizer calculation for softmax. https://arxiv.org/abs/1805.02867
> > >
> > > [3] FlashAttention: Fast and Memory-Efficient Exact Attention with IO-Awareness. https://arxiv.org/abs/2205.14135

---

> > > > ### Author Response · Authors · 2023-11-21
> > > > **Response to reviewer dMuy (continued)**
> > > >
> > > > We would like to further highlight a second significant contribution of our paper: we present the first set of experiments using full attention trained on a massive context length, demonstrating good performance across several machine learning tasks.
> > > >
> > > > We demonstrate that Ring Attention enables the training of the Vicuna chatbot to a 512K context size with just 32 GPUs. In contrast, prior state-of-the-art methods based on memory-efficient or flash attention can only handle a 16K context size. This represents a 32-fold increase in context window size compared to the previous long-context Vicuna. The model's performance is competitive with GPT-4 and Claude-2 in long-context line retrieval tasks and can process sequences over five times longer than those models.
> > > >
> > > > Furthermore, we have applied Ring Attention to scale agentic transformers across hundreds of hindsight-relabeled trajectories with a context size exceeding several million. This approach has led to substantially improved in-context reinforcement learning (RL) performance, achieving state-of-the-art results in these tasks.
> > > >
> > > > To the best of our knowledge, nobody has previously done experiments with full attention at this size of Transformers before, and our success has clear downstream implications for other ML work, such as all the work in RAG.

---

### Official Review · Reviewer_haGA · 2023-11-02

**Soundness:** 3 good
**Presentation:** 3 good
**Contribution:** 3 good
**Rating:** 6
**Confidence:** 4

**Summary:**

This paper proposed an efficient distributed computation of full attention mechanism of long sequences across the sequence dimension. By leveraging the blockwise computation of full attention, the proposed Ring Attention distributes long sequences across multiple devices while overlapping the communication of key-value blocks with the computation of blockwise attention. Importantly, Ring Attention enables us to store the output of each layer across multiple devices, significantly reduces the memory demand in Transformer for long sequences.

**Strengths:**

The challenge the paper aims to address is well-motivated and the proposed method is well-introduced. The experimental results are strong.

**Weaknesses:**

There are concerns/questions about the proposed Ring Attention:

1. If I understand correctly, the Ring Attention is based on the assumption that the blockwise computation of self-attention on different devices is well-balanced. However, in the causal mode of self-attention, which is widely used in auto-regressive LLMs, this assumption does NOT hold. It is because under the causal mode, only the lower triangular blocks are necessarily to compute, yielding around half of FLOPs of the non-causal mode with full attention. But in Ring Attention, even under causal mode, we still need to compute all the blocks in the full attention matrix. When the sequence is super long, this may require significantly more FLOPs?

2. For the results in Table 3, Ring Attention are not compared with other sequence parallel mechanism, such as Deep Speed Ulysses which leverages all-to-all communication to speed up sequence parallelism.

**Questions:**

NA.

---

> ### Author Response · Authors · 2023-11-20
> **Response to reviewer haGA**
>
> Dear Reviewer haGA,
>
> Thank you so much for the positive review and for highlighting that our paper is well-motivated, the proposed method is well-introduced, and for acknowledging our strong experimental results. We appreciate your feedback and suggestions, which we have incorporated into the revised version to improve it.
>
>
> **#1: Comparison with Deep Speed Ulysses.**
>
> We appreciate the pointer to Deep Speed Ulysses which appeared on Arxiv 3 days before the ICLR deadline.
>
> The idea of Deep Speed Ulysses involves combining sequence parallelism and tensor parallelism to leverage their optimized all-to-all NVSwitch-based topology. First, it splits along the sequence dimension with sequence parallelism. Then, before computing attention, it aggregates query, key, and value using all-to-all to ensure that each host has the complete sequence; each host processes one attention head for tensor parallelism of attention scores. Finally, it uses all-to-all again to collect results along attention heads and reshards along the sequence dimension. This method helps in reducing the communication cost of sequence parallelism.
>
> Importantly, Deep Speed Ulysses requires gathering and storing the entire sequence on each device, which becomes an issue for large context windows. In contrast, Ring Attention distributes sequences across devices and communicates key-value pairs in a ring that no device ever holds the entire sequence. Therefore, Ring Attention can scale context size linearly with device count, whereas Deep Speed Ulysses cannot.
>
> Deep Speed Ulysses is great for leveraging all-to-all topology but all-to-all is difficult to scale to a large number of devices. For instance, for H100, the maximum number of GPUs supported with nvlink all-to-all interconnect is limited by network switch. In contrast, TPU torus design can maintain a fast interconnect arbitrarily, such as on thousands of TPUs with optical circuit switching. Since Ring Attention requires a minimal ring topology, it works great on both GPU and TPU, and scales particularly well to large TPU pods. For instance, Ring Attention allows a >10M context size for LLaMa LLM with TPU 1024, which is 512 times longer than the prior state-of-the-art in memory efficiency/flash attention.
>
> We conducted a comparison in end-to-end training of LLaMa (7B, 13B, 34B) using 512x A100 80GB. We compared the maximum context size achieved by Deep Speed Ulysses and Ring Attention, and measured their model flops utilization (MFU). The table below shows the results. Ring Attention not only achieves 64 times longer context size thanks to distributing the sequence across devices but also attains much higher MFU due to overlapping communication in a ring of devices. It’s worth noting that self-attention has lower MFU than feedforward network, and larger context can lower MFU. Even so, Ring Attention outperforms the baseline, showing its effectiveness.
>
>
> | | Context size (x1e3) | | MFU(%) | |
>
> |---------|---------------------|-------|-------|----|
>
> | | Ulysses | RingAttention | Ulysses | RingAttention |
>
> | 7B | 256 | 16384 | 38 | 42 |
>
> | 13B | 128 | 8192 | 40 | 44 |
>
> | 34B | 64 | 4096 | 41 | 45 |
>
>
>
> **#2: Saving compute in casual mode.**
>
> This is a great, insightful suggestion for Ring Attention. By skipping the computation of upper triangular blocks and balancing the computation load between devices when using causal attention, the compute cost can be reduced, potentially leading to an increase in speed. We are definitely interested in this and plan to research this direction in future work.
>
>
> Please let us know if our response resolves your concerns. We look forward to hearing from you. Thank you so much.

---

### Meta-Review · Area_Chair_rDhK · 2023-12-13

**Metareview:**

This is a borderline paper with a disagreement between reviewers. On one hand reviewers agree that paper considers a simple, well known ideas to scale attention computation to longer contexts (512k), which is an impressive scaling to achieve. On the other hand, since most of the ideas are well known, reviewer dMuy is arguing for rejection on the basis of limited novelty. While I agree with reviewer dMuy about limited novelty, I think the paper takes a step towards an interesting direction in scaling attention context length, with open-source code this can help the community advance in this direction. So I am suggesting acceptance.

**Justification For Why Not Higher Score:**

limited novelty

**Justification For Why Not Lower Score:**

Interesting results scaling attention context length to 512k.

---

### Decision · Program_Chairs · 2024-01-16

Accept (poster)